



# Comparison of noise levels of different magnetometer types and space environments

Gerlinde Timmermann[1], David Fischer[2], Hans-Ulrich Auster[1], Ingo Richter[1], Benjamin Grison[3], and Ferdinand Plaschke[1]

[1]Institute of Geophysics and Extraterrestrial Physics, TU Braunschweig, Braunschweig, Germany
[2]Space Research Institute, Austrian Academy of Sciences, Graz, Austria
[3]Institute of Atmospheric Physics of the Czech Academy of Sciences, Department of Space Physics, Prague, Czech Republic

**Correspondence:** Gerlinde Timmermann (gerlinde.timmermann@tu-braunschweig.de)

**Abstract.** The plasma environment around Earth has markedly different characteristics of the magnetic field across distinct spatial regions. In the solar wind, beyond Earth's magnetic influence, the magnetic field is relatively low and less fluctuating. In contrast, the magnetosheath — the region between the bow shock and the magnetopause — is characterized by significantly more turbulent magnetic fields. Within the magnetosphere, the magnetic field can go up to tens of thousands of nanotesla (nT). Traditionally, fluxgate magnetometers have been the standard instrument for space-based magnetic field measurements. However, in recent years, alternative technologies such as anisotropic magnetoresistive (AMR) sensors and optically pumped magnetometers have been proposed and, in some cases, deployed. This study compares the noise performances of two magnetometers, a fluxgate and an AMR, by evaluating their amplitude spectral density measurements across various near–Earth regions of space. The potential of each sensor type for investigating specific phenomena is also evaluated.

## 1 Introduction

In space plasma physics, the knowledge of the magnetic field is of primary importance. Space plasmas are almost always magnetized and the plasma dynamics are significantly dependent on the ambient fields via the Lorentz force. Consequently, measuring the magnetic field in space has been a primary goal since the beginning of the space era. Therefore, fluxgate magnetometers (FGM) have usually been used due to their small size, mass, and power consumption paired with robustness and measurement accuracy. Early examples of scientific space missions carrying FGMs are the Explorer 6 mission from 1959 (Judge and Coleman Jr., 1962) and the Voyager 1 and 2 missions, both launched in 1977 (Behannon et al., 1977). The Pioneer Venus Orbiter mission launched into space in 1978 had a FGM on board as well (Russell et al., 1980). Newer multi–spacecraft missions carrying FGMs around the Earth's magnetosphere are the Cluster mission (Balogh et al., 2001), the Time–History of Events and Macroscale Interactions during Substorms (THEMIS) mission (Angelopoulos, 2008; Auster et al., 2008) and the Magnetospheric Multi-Scale (MMS) mission (Burch et al., 2016; Torbert et al., 2016). FGMs are also on their way to Jupiter on the JUICE (Jupiter Icy Moons Explorer) mission (Brown, 2024) and flying to Mercury onboard BepiColombo (Heyner et al., 2021). The diversity of these missions and their profiles, exploring planetary magnetospheres, measuring the solar wind, and even venturing into interstellar space, shows the versatility of FGM instruments.





However, magnetometers of fluxgate type are not the only ones being used on scientific spacecraft. Over the last decades,
different magnetometers have been flown on missions to various parts of the solar system. Anisotropic magnetoresistive (AMR)
magnetometers were used on the TRIO CINEMA mission (Brown et al., 2012, 2014) in low Earth orbit (LEO) for space weather
investigations. The benefit of such magnetometers is their low weight and robustness, whereas the offset stability and overall
measurement accuracy is diminished compared to scientific FGMs (Schulz et al., 2019). Another example for the applicaton of
AMR magnetometers is the so–called Service Oriented Space Magnetometer (SOSMAG); it has been part of the Korean GEO–
KOMPSAT–2A spacecraft launched in 2018 into geostationary orbit (Magnes et al., 2020). SOSMAG consists of two AMRs
inside the spacecraft and two FGMs mounted on a one meter long boom, which is very short in comparison to the spacecraft
dimensions (Leitner et al., 2015). Here, the AMR magnetometers are used to measure high intensity magnetic disturbances
generated inside the spacecraft in order to be able to correct for them in the more precise fluxgate measurements.

An Overhauser magnetometer (or proton precession magnetometer) was used in the Danish Ørsted mission to newly measure
Earth's magnetic field (Neubert et al., 2001). This kind of magnetometer features high accuracy and stability. However, it can
only perform scalar measurements, the sampling rate is low, requires a high ambient field to operate at all, and is rather heavy
in comparison to AMR and fluxgate magnetometers.

Another group of magnetometers falls into the category of optically pumped magnetometers, of which the vector helium
magnetometer (VHM) is a flight–proven example. The Ulysses mission launched in 1990 used a VHM together with a flux-
gate magnetometer (Balogh et al., 1992). The VHM combines a high offset stability with the ability to measure the absolute
magnetic field. Thus, the VHM can be used to calibrate the fluxgate magnetometer with very high accuracy, also deep within
planetary magnetospheres. The same measurement principle was also applied in the Cassini mission targeting the Saturnian
system (Dougherty et al., 2004). VHMs are also flying on the three–satellite mission Swarm in LEO to study Earth's magnetic
field (Leger et al., 2009). Here, vector fluxgate magnetometers were also used in combination on each satellite.

The coupled dark state magnetometer (CDSM) (Magnes et al., 2013) is a new optically pumped magnetometer that is part
of the JUICE mission on its way to Jupiter. Furthermore, it has been launched on the China Seismo–Electromagnetic Satellite
(CSES) and is now investigating natural electromagnetic phenomena in LEO (Pollinger et al., 2018). The CDSM is a scalar
magnetometer that is able to measure the magnetic field modulus irrespective of the field direction with respect to the sensor
cell. It measures the magnetic field using the Zeeman effect of rubidium in a small glass cell filled with an additional buffer gas.
(Pollinger et al., 2018) The shifting and splitting of the hyperfine structure energy levels as a function of the ambient magnetic
field is combined with coherent population trapping. This leads to narrow resonance features and also enables omni–directional
measurement. Other magnetometers are also under development, for example on the basis of nitrogen vacancies in diamond
(Stürner et al., 2019; Webb et al., 2019), though these are not space–ready yet (Bennett et al., 2021).

As stated, the various magnetometers all have different characteristics that can be compared, among others: noise (frequency
dependent), ability to perform scalar vs. vector measurements, stability with respect to calibration parameters (offsets, gains,
direction of magnetic axes) and their linearity, system complexity, robustness, power consumption, price, and size. Depending
on the mission or spacecraft constraints, the mission's scientific objectives and duration, and the visited space region's charac-
teristics, a specific magnetometer type has to be chosen. While FGMs have been the traditonal choice, it shall be investigated




whether other types might also be suitable for certain regions or conditions. This work is a first step to systemetically look
for required and realized noise levels in near–Earth space environments. All of the magnetometers mentioned above measure
space phenomena in the frequency range from DC to tens of Hz. For higher frequencies search coil magnetometers are used
(e.g., Roux et al., 2008; Le Contel et al., 2016). For a comparison of search coil vs. fluxgate magnetometers and a merging of
their observations, see Fischer et al. (2016). It is important to note that the measurement performance of the satellite missions
considered is not limited by the magnetometers themselves, but rather by the magnetic properties of the satellite. The satellites
generate significant DC and AC magnetic disturbances –— particularly evident in later sections —– which are detected by
magnetometers mounted on short booms. Consequently, ensuring magnetic cleanliness, especially in the AC regime, is more
critical than further improvements in instrument sensitivity or accuracy.

In this paper we investigate the noise levels of FGMs and AMR magnetometers in a magnetically quiet environment. These
are then compared to fluxgate measurements from geostationary orbit (GEO) inside the Earth's magnetosphere (from SOS-
MAG), from the magnetosheath and from the pristine solar wind upstream (both from THEMIS). Data from geostationary orbit
represent mostly undisturbed measurements inside the magnetosphere. To contrast these, we also included Cluster measure-
ments from a polar orbit taken during times of electromagnetic ion cyclotron (EMIC) wave activity. These are common waves
in the magnetosphere that are produced by anisotropic proton distributions in the $0.1$ to $5$ Hz frequency range. They are usually
transverse left–hand polarized waves (Usanova et al., 2012).

## 2   Data and Methods

In this paper, six data sets are investigated: (a) fluxgate magnetic field measurements in a magnetically quiet laboratory environ-
ment, (b) AMR measurements also in a magnetically quiet laboratory environment, (c) Cluster FGM measurements of EMIC
waves in Earth's magnetosphere, (d) SOSMAG measurements in GEO, (e) THEMIS FGM measurements in the terrestrial
magnetosheath, and (f) THEMIS FGM measurements in the solar wind:

(a) A science grade FGM similar to the ones used on Rosetta (see Auster et al., 2007), THEMIS (see Auster et al., 2008),
and JUICE (see Brown, 2024) is put in a magnetically shielded environment on the ground to measure the magnetic field
overnight with a data rate of $4$ Hz. The whole data series is 9 hours and 12 seconds long. In order to remove disturbing
trends from external effects at the beginning and the end, only data between 5000 s and 25,000 seconds (about 83 min
to 7 h) (Auster and Timmermann, 2025).

(b) An AMR magnetometer as used in SOSMAG (Leitner et al., 2015; Magnes et al., 2020) is put in a shielded environment
on the ground overnight and the magnetic field is measured with a data rate of $2$ Hz. The whole data series is 16 hours,
24 minutes, and 27 seconds long (Valavanoglou and Timmermann, 2025).

(c) The EMIC wave data set were selected based on Cluster data of spacecraft 1 between 5 April 2001 and 19 Nov 2018
(B. Grison, personal communication). The list contains $479$ events. Of these events, one is randomly chosen, split into
5 minute intervals and then one interval out of that is randomly selected until a total of 500 intervals are picked (Grison



and Timmermann, 2025). The number of intervals is reduced compared to the other data sets, because at 1000 intervals the EMIC waves were not visible in the analysis anymore (see section 4). The Cluster spacecraft 1 FGM data used are of five vectors per second type, yielding a data rate of 5 Hz. Cluster data are publicly available via the Cluster Science Archive (Laakso et al., 2010), including the Cluster FGM data set (European Space Agency, 2025).

(d) GEO magnetic field measurements were taken from SOSMAG FGM data on board GEO–KOMPSAT–2A at geographic longitude of 128.2° East inside Earth's magnetosphere. The data was measured with a data rate of 1 Hz between 1 March 2019 and 12 Oct 2024 (Magnes et al., 2020). Within that time range 1000 five minute intervals are randomly selected (Timmermann, 2025a).

    (e) Selection of the magnetosheath (MSH) data set is based on the list provided by Koller et al. (2021) of THEMIS magne-
tosheath observations between 24 June 2008 and 31 Dec 2020. First, an interval from the list that is at least five minutes long is randomly picked. Then, a five minute interval out of that is randomly chosen. This process is repeated until there are 1000 intervals selected. THEMIS FGM data of FGL type (fluxgate magnetometer low resolution) were used with a data rate of 4 Hz (Timmermann, 2025b).

    (f) The solar wind (SW) data set was selected based on a list of THEMIS B data (T. Glißmann, personal communication)
that contained 94 entries with varying length (minutes to days). These were further selected by visual inspection of the omni-directional ion energy flux to only include intervals of pristine solar wind (no foreshock) measurements. This led to a list of 1856 five minute intervals taken between 12 June 2008 and 28 Sep 2008. From this list 1000 intervals are randomly selected. Here, only THEMIS B FGM data of FGL type with a data rate of 4 Hz were used (Timmermann, 2025c).

The random picking of 1000 intervals for data sets (d) to (f) and 500 for data set (c) has been chosen to get statistical significance while limiting the computation time. Additionally, this leads to a noise floor where all disturbances are from the spacecraft (as seen later) and only general space phenomena are perceptible. It has been checked in several runs for each data set that the overall results do not depend on the exact combination of chosen intervals. For the selected intervals of data sets (c) to (f), the average power spectral density (PSD) per component is calculated with a discrete Fourier transform according to:

$$P(f_k) = \frac{1}{3}\sum_{xyz}\frac{2\,\Delta t}{N}\Big|\sum_{m=0}^{N-1}B_{xyz,m}\exp(-2\pi i\frac{mk}{N})\Big|^2 \qquad (1)$$

where $P(f_k)$ is the PSD at frequency $f_k = k/(N\,\Delta t)$, $\Delta t$ denotes the sampling period, $N$ is the number of sampling points per interval and $B_{xyz,m}$ represents the magnetic field measurement observation in one component ($x$, $y$, or $z$) at position $m$ within the interval. For each data set and frequency, the 5th, 10th, 25th, 50th, 75th, 90th, and 95th percentiles of the square roots of the PSDs ($P(f)^{1/2}$, also called the amplitude spectral density) are calculated. This is done to see the range and will be
quantified in section 3. For the two magnetometers (data sets (a) and (b)), the average PSD of the whole data set is computed using Welch's method (Welch, 1967; SciPy, 2025). First, the data set is split into intervals of a length of 4000 (FGM) and 2000





**Table 1.** Amplitude spectral density P(f)$^{1/2}$ of the 50th quantile of the space data sets at 3 mHz, 0.1 Hz, and 1 Hz (for GEO data at 0.5 Hz). For the lab data sets, the amplitude spectral density is given at 1 mHz, 0.1 Hz, and 1 Hz.

| Data set | P(f)$^{1/2}$ @ 3 mHz (1 mHz) | P(f)$^{1/2}$ @ 0.1 Hz | P(f)$^{1/2}$ @ 1 Hz (0.5 Hz) |
|---|---|---|---|
| FGM | 77.91 pT Hz$^{-1/2}$ | 21.43 pT Hz$^{-1/2}$ | 9.36 pT Hz$^{-1/2}$ |
| AMR | 1277.36 pT Hz$^{-1/2}$ | 282.41 pT Hz$^{-1/2}$ | 157.09 pT Hz$^{-1/2}$ |
| EMIC | 13578.00 pT Hz$^{-1/2}$ | 377.20 pT Hz$^{-1/2}$ | 49.19 pT Hz$^{-1/2}$ |
| GEO | 3843.12 pT Hz$^{-1/2}$ | 155.01 pT Hz$^{-1/2}$ | 52.25 pT Hz$^{-1/2}$ |
| MSH | 47409.72 pT Hz$^{-1/2}$ | 5233.36 pT Hz$^{-1/2}$ | 318.50 pT Hz$^{-1/2}$ |
| SW | 3184.24 pT Hz$^{-1/2}$ | 197.23 pT Hz$^{-1/2}$ | 28.28 pT Hz$^{-1/2}$ |

(AMR) points, then the overlap is defined to be half of the segment length. The difference in the number of points is due to the different data rates of the magnetometers, it leads to a resolution of 1 mHz in both cases. Second, the data is detrended linearly. Third, the overlapping data intervals are windowed with a Hamming window. Fourth, the discrete Fourier transform is calculated for each window according to Eq. 1. The averaging of the calculated periodograms is done by using the mean. This is done for each component of the magnetic field indiviudally, then they are all summed up and divided by three to get the amplitude spectral density per component.

## 3 Results

Results pertaining to the data sets (a) through (f) are shown in the corresponding panels (a) through (f) of Fig. 1. The square roots of the PSDs of each of the random intervals is calculated and then, the percentiles thereof are shown in Fig. 1. Note that the axis scalings are different for the lab measurements and space measurements.

The values of the 50th percentiles of the amplitude spectral densities P(f)$^{1/2}$ of each data set at 3 mHz, 0.1 Hz, and 1 Hz (for GEO at 0.5 Hz) as shown in panels (a) through (f) of Fig. 1 are listed in Table 1. For the FGM and AMR data sets, the values are given at 1 mHz, 0.1 Hz, and 1 Hz. The values at 0.1 Hz are included as an extra point of comparison, because the standard point of comparison at 1 Hz used in literature is in this case quite near to the Nyquist frequency. This might lead to effects on the measured values due to the internal filters of the instrument.

All the panels in Fig. 1 show a downward slope of the spectrum. The spectral slope $\alpha$ in the log–log scale is given by

$$\alpha = \frac{\log \frac{P(f_2)}{nT^2/Hz} - \log \frac{P(f_1)}{nT^2/Hz}}{\log \frac{f_2}{Hz} - \log \frac{f_1}{Hz}} = \frac{\log \frac{P(f_2)}{P(f_1)}}{\log \frac{f_2}{f_1}} \tag{2}$$

where $P$ is the value of the spectrum, $f$ is the frequency and the indices 1 and 2 denote two different points on the spectrum. The spectral slope is often used as a measure of turbulence. (Alexandrova et al., 2012; Borovsky, 2012) All the spectral slopes $\alpha$ are listed in Table 2. Note that the slope is calculated without taking the square root of the spectrum. Therefore, the slopes taken from the spectra (e.g., in Fig. 1) must be multiplied by 2 to match the values listed in Table 2. For the AMR and the magnetosheath data sets, there are two slopes. The first slope of the AMR data set is calculated from the second data point until





**Figure 1.** Overview of percentiles of amplitude spectral densities of all space regions and magnetometer data considered. Note that the axis scalings are different for the magnetometer panels compared to the rest. The legend in panel (f) is valid for all panels (c) through (f). For better readability, the 50th percentile line is a bit thicker than the others.





**Table 2.** Slopes of the 50th percentile of all data sets and variations of data sets (c) to (f). For the AMR data set, the first slope was calculated between 1 mHz and 0.03 Hz and the second between 0.1 Hz and 0.999 Hz. For the MSH data set, the corner frequency between the two regions was set at 0.2 Hz. For all other data sets, the second and the second to last data points were used (see Fig. 1). Note that the slope is calculated without taking the square root of the spectrum. Thus, the slopes taken from the spectra in Fig. 1 need to be multiplied by 2 to obtain the values listed in the table below.

| Data set | Slope $\alpha$ | variation $\mathcal{V}$ |
|:---:|:---:|:---:|
| FGM | $-0.75$ | |
| AMR | $-1.02$ | |
| | $-0.23$ | |
| EMIC | $-1.81$ | 1.20 |
| GEO | $-1.72$ | 0.80 |
| MSH | $-1.37$ | 1.11 |
| | $-2.56$ | |
| SW | $-1.68$ | 0.78 |

$3 \cdot 10^{-2}$ Hz. The second slope was calculated between 0.1 and 1 Hz. For the magnetosheath data set, the corner frequency of 0.2 Hz was used. There are two distinct regions in the corresponding panels, as can be seen in Fig. 1(b) and (e). For all other data sets, the second and the second to last data points were used.

A second characteristic is the variation $\mathcal{V}$ of $P(f)^{1/2}$ of the data sets, which is given by

$$\mathcal{V} = \frac{1}{N/2} \sum_{k=1}^{N/2} \left( \log \frac{P_{95}^{1/2}(f_k)}{\text{nT/Hz}^{1/2}} - \log \frac{P_5^{1/2}(f_k)}{\text{nT/Hz}^{1/2}} \right) = \frac{1}{N/2} \sum_{k=1}^{N/2} \log \frac{P_{95}^{1/2}(f_k)}{P_5^{1/2}(f_k)} \tag{3}$$

where N is the number of sampling points, $P_{95}$ is the 95th percentile of PSDs, $P_5$ is the 5th percentile of PSDs, and $f_k$ is the frequency. The variation is a measure for the variability of the data set: the larger the variation, the more different in spectral power the individual intervals are. The variation of the percentiles of the data sets is also listed in Table 2.

With slope and variation defined, we can now evaluate the data sets. The least steep slope over a complete data set can be seen in Fig. 1(a), where the FGM data set is shown. The next higher slope is exhibited by the amplitude spectral density of the AMR shown in Fig. 1(b) with at first $\alpha = -1.02$. It can be seen to flatten at $3 \cdot 10^{-2}$ Hz to $\alpha = -0.23$ (see Table 2). At low frequencies, the overall values of the amplitude spectral densities are one order of magnitude higher for the AMR compared to the FGM. At 0.5 Hz the amplitude spectral density of the AMR is a factor of more than 12 higher than the FGM's (see Table 1).

The GEO data set shown in Fig. 1(d) exhibits an average slope of $\alpha = -1.72$ with values of the amplitude spectral density between $3843 \, \text{pT} \, \text{Hz}^{-1/2}$ at 3 mHz and $52 \, \text{pT} \, \text{Hz}^{-1/2}$ at 0.5 Hz.

The EMIC wave data set shown in Fig. 1(c) behaves similarly to the GEO data set collected in Earth's magnetosphere. The spectrum has higher values of the amplitude spectral density at low frequencies (see Table 1). It exhibits a slightly higher slope of $\alpha = -1.81$ than GEO that leads to higher values of the amplitude spectral density at 0.5 Hz, but continues to decrease to 49 $\text{pT} \, \text{Hz}^{-1/2}$ at 1 Hz. The amplitude spectral density of the EMIC wave data set also features a slight bump above 0.2 Hz. The





magnetosheath data set featured in Fig. 1(e) shows a less steep slope of $\alpha = -1.37$ compared to EMIC and GEO in the lower frequencies up to 0.2 Hz. A break in the slope is observed at this point, with the slope value of $\alpha = -2.56$ being the steepest among all data sets (see Table 2). In the last panel of Fig. 1, the solar wind data set is shown with a slope of $\alpha = -1.68$.

The variation $\mathcal{V}$ of the data sets can only be given for data sets (c) through (f) of Figure 1, since only for those several amplitude spectral densities were available. The biggest variation $\mathcal{V}$ with 1.2 can be seen in the EMIC data set in Fig. 1(c). The magnetosheath data set in Fig. 1(e) also exhibits a large variation with $\mathcal{V} = 1.11$. The GEO and SW data sets in Panels (d) and (f) have very similar, smaller variations with $\mathcal{V} = 0.8$ and $\mathcal{V} = 0.78$, respectively. Some of the data sets also exhibit distinct spikes in their PSDs. In the AMR data set in Fig. 1(b), those are at 0.25 Hz and 0.5 Hz caused by housekeeping transmission. The EMIC data set in Fig. 1(c) has a spike at 0.24 Hz which is close to the spin frequency of 0.25 Hz of the spacecraft (Cluster has a spin period of 4s (Balogh et al., 2001)). In Figure 1(d), a spike at 0.166 Hz is visible in the GEO data set. This spike has been attributed to a disturbance source located in the spacecraft (Magnes et al., 2020). In Fig. 1(f), the solar wind data features several spikes at 0.33 Hz and multiples thereof. These can be attributed to the spacecraft's spin frequency and its harmonics.

To assess the suitability of the two magnetometer types to make sensible measurements in the respective regions of space, we have to compare the maximum noise floor of their lab environment spectra with the lowest PSD values (5% percentiles) of the space spectra. In Fig. 2 exactly this comparison is shown. The spectra pertaining to the magnetometers are plotted in light and dark blue; the spectra pertaining to the other data sets are depicted in other colors. Plots likes this with the 50th and 95th percentiles for the space data sets are available in the appendix. If the magnetometer's spectra are below those of the space environments, this means it is possible to use the magnetometer to measure all of the phenomena of interest in those environments. If half of the spectra of the space environments were higher than the magnetometer's spectra, half of the phenomena would be invisible. It can be seen that the AMR magnetometer can only be used to measure in regions with higher values of the amplitude spectral density, i.e. continuously above $200 \text{ pT Hz}^{-1/2}$. This is for example given in the magnetosheath until 0.5 Hz. In all other regions, at frequencies up to a maximum of $2.5 \cdot 10^{-2}$ Hz, the AMR noise floor is above the phenomena of interest. In contrast, the fluxgate spectrum remains below, or at the same level as, the others across the entire frequency range.

## 4  Discussion

Turbulence describes the process of energy transport between varying magnetic fields and motions from larger to smaller spatial scales. (Alexandrova et al., 2013; Klein et al., 2023) The spectral slope $\alpha$ is an important feature of turbulent plasmas. It changes in distinct regions of frequencies. (Alexandrova et al., 2012) In Fig. 1, the amplitude spectral densities of all data sets are shown. Since all the spectra were obtained from space plasmas, they generally follow Kolmogorov's law for turbulent spectra $f^{-5/3}$ (Kolmogorov, 1941). Comparing the slope of the solar wind data set shown in Fig. 1(f), which in our case was at $\alpha = -1.68$ (see Table 2), it is very similar to other values reported in literature. In Borovsky (2012), the reported spectral slope (sometimes also called spectral index) of the solar wind is $\alpha = -1.62$, in Alexandrova et al. (2009) it is at $\alpha = -1.7$ for the respective frequency range of $10^{-3}$ to 1 Hz, and in Roberts et al. (2024), it is reported at $\alpha = -1.49$ from $10^{-3}$ to 0.2





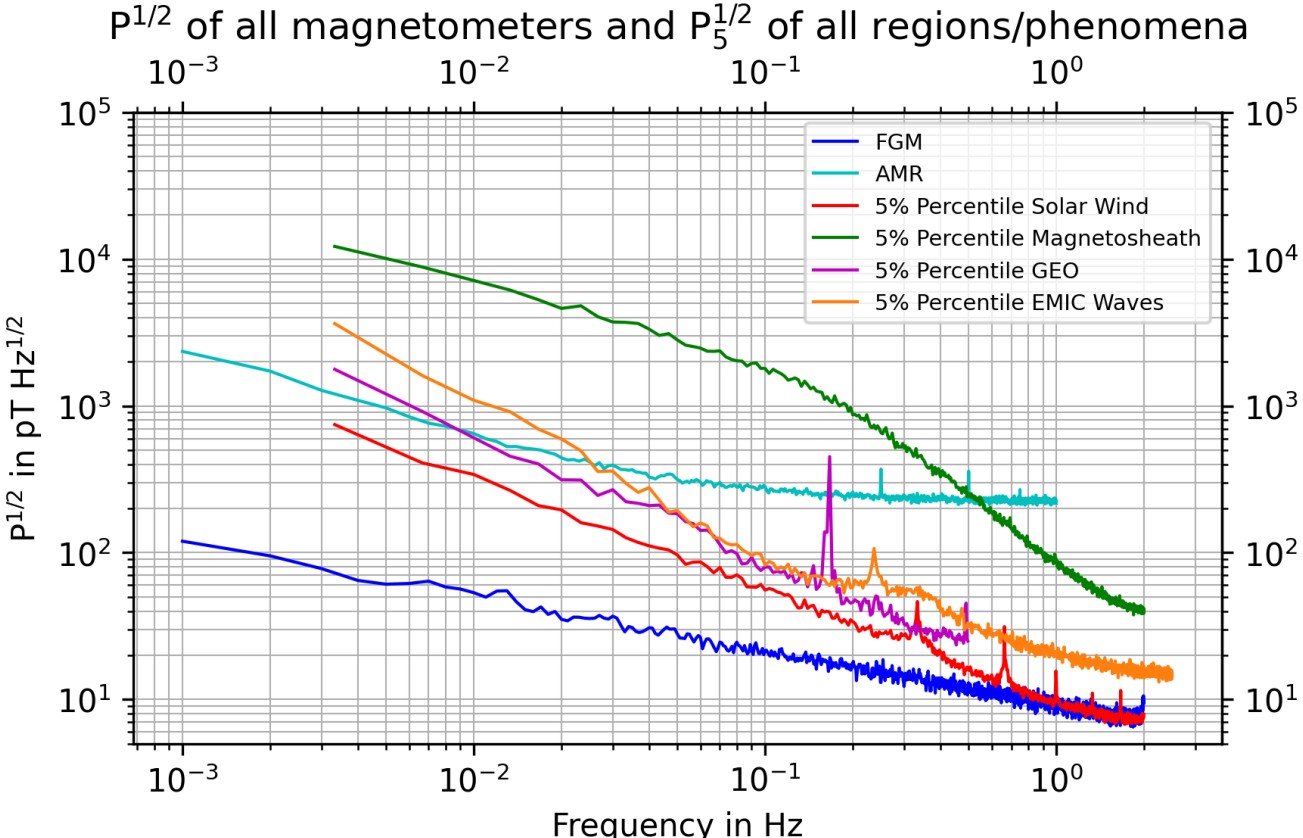

**Figure 2.** The plot compares $P_5^{1/2}$ of the regions and phenomena with $P^{1/2}$ of the magnetometers. The differences in amplitude spectrum length are due to different data rates of the sensors used. Plots likes this with the 50th and 95th percentiles for the space data sets are available in the appendix.

Hz. Compared to the other data sets of regions and phenomena included in this paper, the solar wind data set has the lowest amplitude spectral density. It includes only pristine upstream solar wind (no foreshock) with structures continually evolving in the plasma on small scales. Other phenomena adding to the spectrum could be e.g. corotating and stream interaction regions, or mesoscale structures (Viall et al., 2021; Rakhmanova et al., 2023).

200     Two of the data sets were collected inside Earth's magnetosphere: the GEO dataset and the EMIC data set. As described in the last section, the EMIC data set has higher values of the amplitude spectral density than the GEO data set. This might be due to the fact that for that data set, specifically times with wave activity were included, whereas for the GEO data set calm times with low wave activity are predominantly present in the data set. Both panels (c) and (d) of Fig. 1 show a spike: for EMIC it is due to the spin frequency of the spacecraft, for GEO the spike is due to a disturbance as explained in the last section.

205 In the EMIC data set, a bump appears between 0.2 Hz and 1 Hz. This bump can be attributed to the EMIC waves: in Earth's



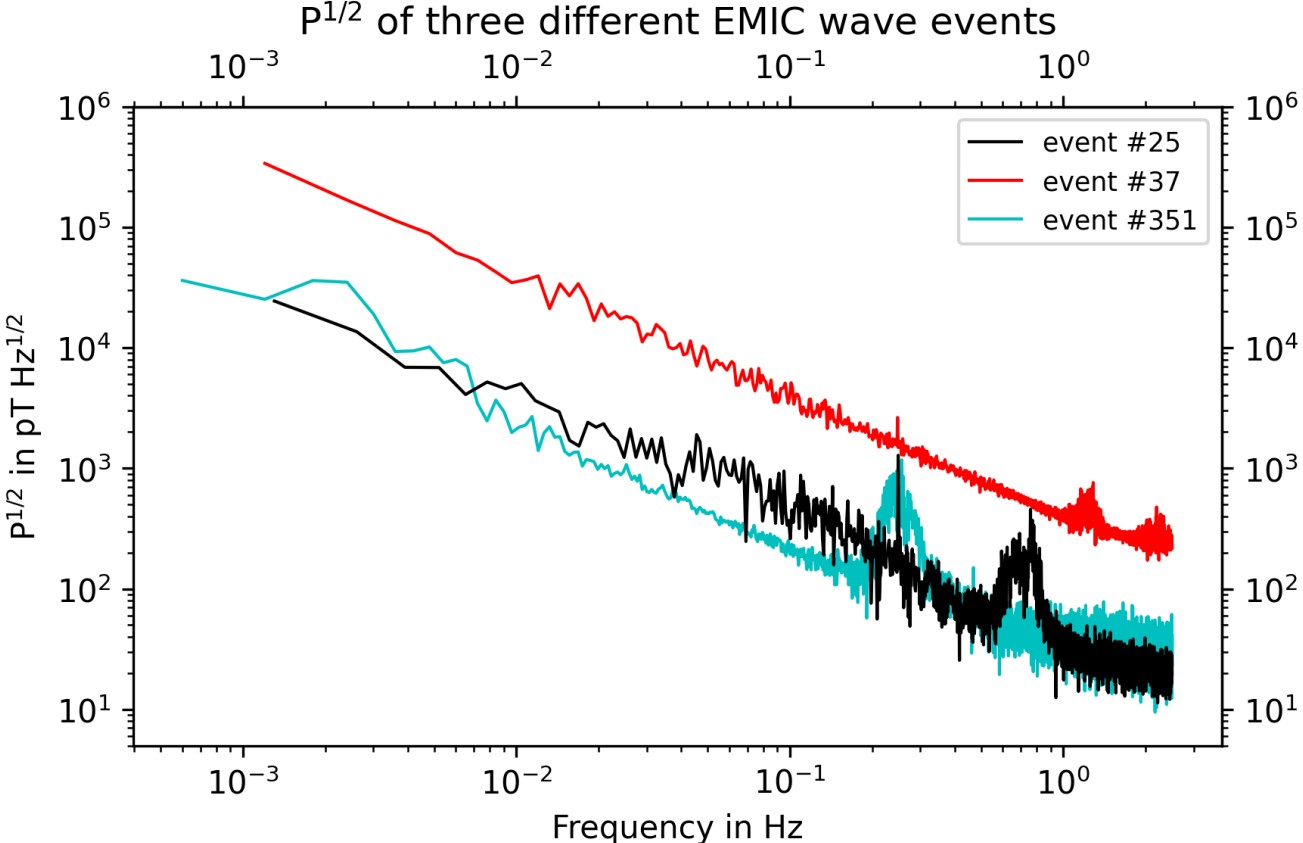

**Figure 3.** The plot shows $P^{1/2}$ of three different EMIC wave events from the data set. The events differ in the size and number of the EMIC spike(s) and the frequency it occurs at. This illustrates why there is only a bump in Fig. 1(c): the individual events are in a wider range of frequencies and average each other out. Note that the different background levels are due to different range modes used: event #25 used mode 2, event #351 used mode 3 (both low background levels), and event #37 used mode 4 (higher background level). Each range mode has a different digital resolution.

magnetosphere, these narrowband (few 100s mHz) emissions can occur in a wider frequency range (0.1 to 2.5 Hz in the present study) (Usanova et al., 2012; Grison et al., 2021). The amplitude spectral densities of three single EMIC events are shown in Fig. 3. They have one or more distinct peaks, but at different frequencies in the cited range. This holds true for all the individual events. When all events are averaged, the individual peaks are smoothed in the overall spectrum, as they are averaged with the flat portions of the spectra from other events. This averaging effect leads to the observed bump within the overall frequency range of EMIC wave occurrences. Note that the different background levels are due to different range modes used: event #25 used mode 2, event #351 used mode 3 (both low background levels), and event #37 used mode 4 (higher background level). Each range mode has a different digital resolution.





The magnetosheath data set shown in Fig. 1(e) features a spectral break at $0.2\,\mathrm{Hz}$, which is in the transition region of inertial ranges between ion and electron scales (Alexandrova et al., 2013; Klein et al., 2023). Below $0.2\,\mathrm{Hz}$, the spectral slope was less than Kolmogorov's law for turbulent spectra with $\alpha = -1.37$. Above, it has the largest slope exhibited in all the data sets with $\alpha = -2.56$. This is due to a broad enhancement of the spectrum in the frequency range of $3 \cdot 10^{-2}$ to $1\,\mathrm{Hz}$. In this frequency range, several different phenomena may contribute to the higher amplitude spectral density. In the magnetosheath, the turbulence levels depend on the measurement position and on how the interplanetary magnetic field (IMF) is aligned with the upstream bow shock segment: at a quasi–parallel shock, where the angle between IMF and bow shock normal vector is below $45°$, plasma turbulence is strong (Vörös et al., 2017, 2016). Foreshock waves such as 30s waves (Eastwood et al., 2005) and 3s waves (Le et al., 1992; Blanco-Cano et al., 1999) play an important role as well as so-called magnetosheath jets (Plaschke et al., 2018; Pöppelwerth et al., 2024). Short Large Amplitude Magnetic Structures (SLAMS) (Schwartz and Burgess, 1991; Karlsson et al., 2024) also transmit into the magnetosheath at the quasi–parallel bow shock and might add to the enhancement seen in the data set. Mirror modes (Tsurutani et al., 2010; Volwerk et al., 2014) may be locally generated downstream of the quasi–perpendicular shock and there–by contribute to the overall fluctuating levels in this region of space.

The results obtained thus far can now be compared with the FGM and AMR measurements. The amplitude spectral density of the FGM is shown in Fig. 1(a). It has a spectral slope of $\alpha = -0.75$, approximately following a 1/f noise spectrum as is expected in this frequency range (Hooge et al., 1981). It exhibits the lowest values of all data sets, going down to $9\,\mathrm{pT\,Hz^{-1/2}}$ at $1\,\mathrm{Hz}$. This means the FGM is well suited for measurements in all the listed space regions and of all phenomena occuring therein (also see Fig. 2). If a magnetometer with a higher data rate is chosen, it can also measure at higher frequencies; the FGM used in the MMS mission, for instance, can measure up to $128\,\mathrm{Hz}$ (burst mode) although above $16\,\mathrm{Hz}$ for the merged data product the SCM data is used exclusively on MMS (Torbert et al., 2016).

The spectrum of the AMR magnetometer as shown in Fig. 1(b) has a slope of $\alpha = -1.02$ due to 1/f noise, as seen in other AMR magnetometers (Qiu et al., 2018). A spectral break occurs at $0.05\,\mathrm{Hz}$, above $0.1\,\mathrm{Hz}$ the sensor exhibits only frequency independent thermal noise (Leitner et al., 2015; Johnson, 1928). However, the general noise level is significantly higher with $157\,\mathrm{pT\,Hz^{-1/2}}$ at $0.5\,\mathrm{Hz}$ compared to the fluxgate magnetometer as already seen in the last section. Since this is at quite a high level, this AMR magnetometer is not suitable for scientific measurements at lower frequencies though there are other AMRs with lower noise floors as described in Brown et al. (2012). This finding is also well illustrated in Fig. 2: all amplitude spectral densities except for the one pertaining to the MSH data set mostly lie below the AMR's spectrum, thus making measurements with the AMR in these regions not reasonable.

The variation as listed in Table 2 shows that the regions/phenomena with more wave activity and turbulence also exhibit a higher variation in the amplitude spectral densities. This means that the individual intervals are quite diverse, some with high values of the spectrum due to possibly higher wave activity and turbulence, others with lower values of the spectrum that likely have less wave activity and turbulence. The calculated variations are averaged over the whole spectrum. This is why the variation of the solar wind is seemingly not fitting to the plot: the variation at low frequencies is quite different for this data set compared to the variation at higher frequencies.



Overall, as shown in Fig. 2, the lowest spectrum pertains to the solar wind. EMIC and GEO have similar values of their respective amplitude spectral densities, underlining that the data sets were collected in the same region of Earth's magnetosphere.
The MSH data set shows a turbulent magnetic field, which is mirrored in the corresponding higher values of the spectrum going up to $4.7 \cdot 10^4 \,\mathrm{pT}\,\mathrm{Hz}^{-1/2}$ at 3 mHz. The difference of values in the amplitude spectral densities between the lowest and highest frequencies of each data set is more than 2 orders of magnitude for the SW, MSH, GEO, and EMIC data sets, but only 1 order of magnitude for the magnetometer data sets. One also has to keep in mind that the noise levels seen are also depending on the instrument itself and not only the ambient conditions. The digital resolution decreases with increasing ranges of the instrument
(or the range mode used), so the general noise level accordingly increases with increasing range. This was explicitly shown in Fig. 3 for the different range modes in Cluster. Thus, the FGMs have limitations concerning the amount of telemetry (reflected e.g. in ranges). These limits increase instrument noise levels artificially and are also reflected in the statistics used for this work.

## 5   Summary & conclusions

We've compared the amplitude spectral densities of a FGM and an AMR in a shielded lab environment with spectra obtained
in different regions of space (magnetosphere, magnetosheath, solar wind). The spectral slopes are generally higher in space than in shielded environments. The highest amplitude spectral density values are measured in the magnetosheath. In sum, for all of the measured phenomena and regions, a fluxgate magnetometer will be able to measure the fields and resolve the natural fluctuations within a frequency range of 1 mHz to 2.5 Hz. Higher frequencies are achievable with higher data rates, but were not analyzed here. The AMR magnetometer is only suitable for more turbulent regions such as the magnetosheath.
Alternatively, it can be used in combination with a fluxgate magnetometer to clean the latter's data as was done in SOSMAG.

The selection of a magnetometer depends on the expected phenomena to be measured, with an additional margin in measurement capabilities to ensure robustness and accommodate the need to detect unforeseen events. Furthermore, there is an inherent trade–off between selecting a highly sensitive magnetometer for optimal scientific measurements and the associated costs and system complexities. This paper aims to quantify this noise–related trade–off when choosing a magnetometer for a
mission. Based on noise levels and frequency ranges covered, the data sets presented here favor the FGM over the AMR, which exhibits more limited applicability.

However, only two magnetometer types were considered here. As mentioned in the introduction, there are other types of magnetometers that might be suitable depending on the mission. For some applications, e.g. measurements in the solar wind, the stability of offsets (especially over time) or other calibration parameters (Plaschke et al. (2019)) might be of utmost importance.
Future works could also consider mass and size (of the sensor and the electronics), power consumption, complexity, and price. Different choices in magnetometer types might emerge depending on the exact application. Although similar findings are to be expected in other planetary magnetospheres, future works could explicitly compare this as well as other regions in Earth's magnetosphere such as LEO measurements.





*Data availability.* All data sets are available under their respective DOI as given in the references.

**Appendix A**

To make even more comparisons possible between the lab and space measurements, two more figures are added complementing Fig. 2 in section 4. Figure A1 compares the magnetometer's amplitude spectral density with the 50th percentile of the space measurements and Fig. A2 compares the magnetometers to the 95th percentile of the space measurements.

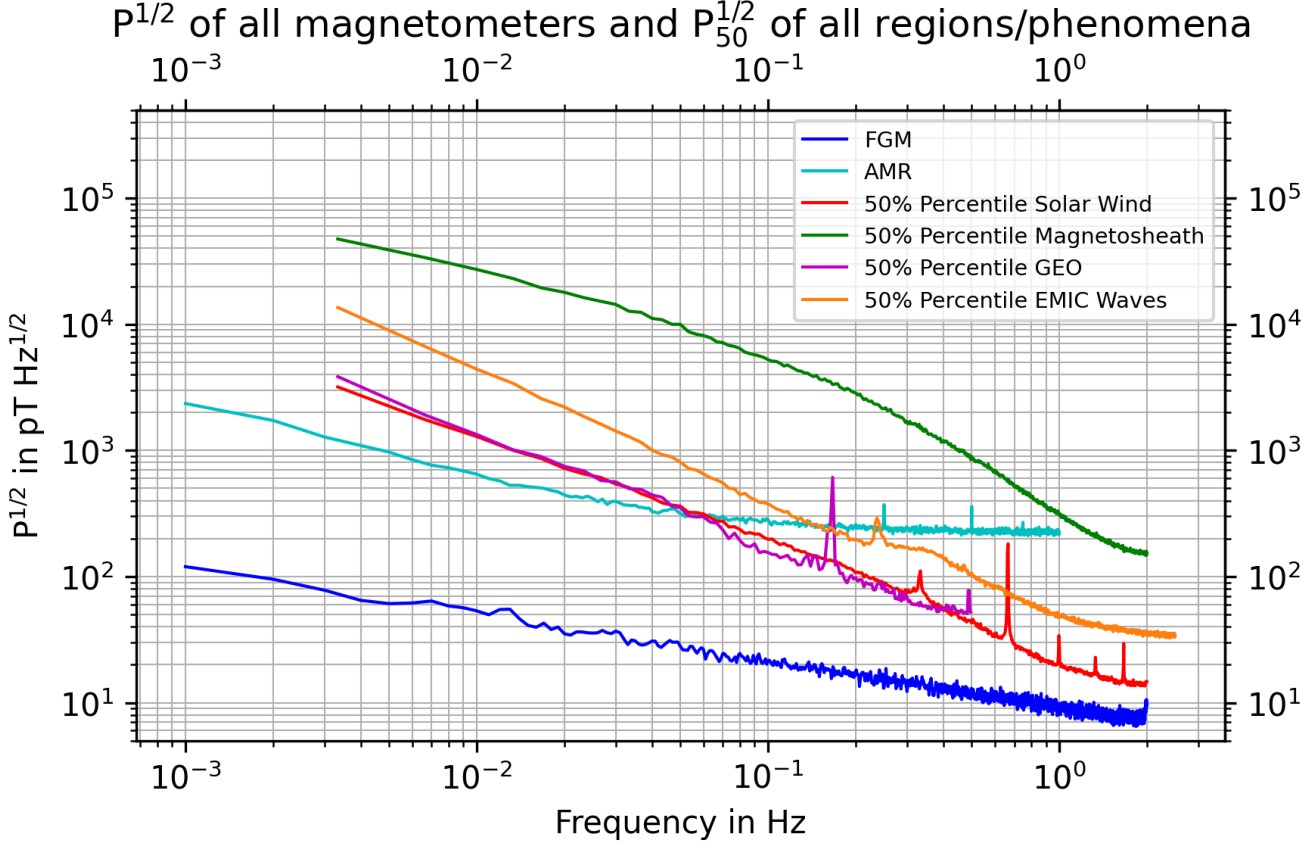

**Figure A1.** The plot compares $P_{50}^{1/2}$ of the regions and phenomena with $P^{1/2}$ of the magnetometers. The differences in amplitude spectrum length are due to different data rates of the sensors used.





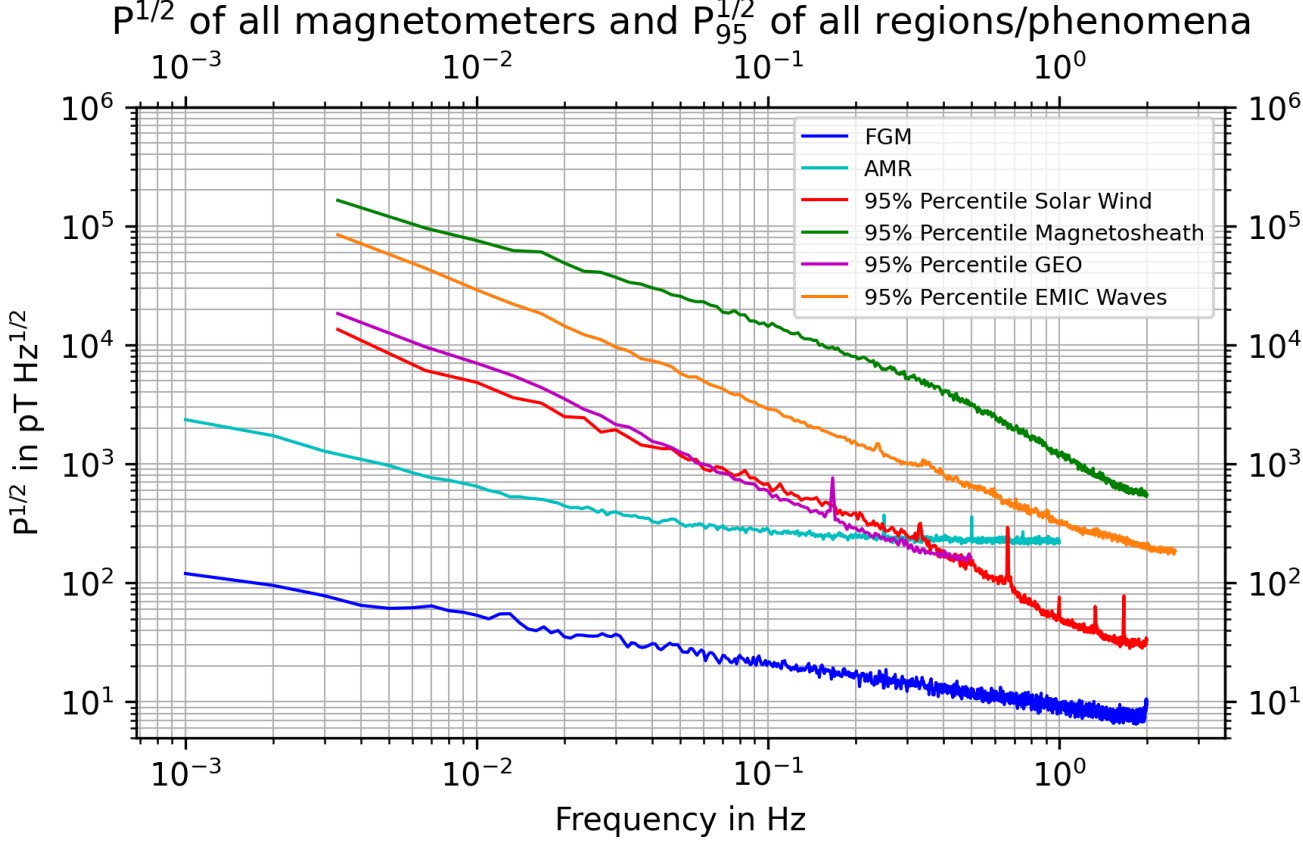

**Figure A2.** The plot compares $P_{95}^{1/2}$ of the regions and phenomena with $P^{1/2}$ of the magnetometers. The differences in amplitude spectrum length are due to different data rates of the sensors used.



*Author contributions.* G. Timmermann prepared the manuscript with contributions from all co–authors. The FGM magnetometer data set
was collected by H.–U. Auster. The EMIC data set was compiled by B. Grison.

*Competing interests.* The authors declare that they have no conflict of interest.

*Acknowledgements.* This work is based on a first study on magnetometer performance done for the QUASIMODO project partly funded by
the German Aerospace Center (DLR) under grant number 50 WM 2171.

Benjamin Grison acknowledges support from the Czech Science Foundation (GA ČR) project 25-19511L (RADIANCE).

THEMIS data can be retrieved from http://themis.ssl.berkeley.edu/data/themis/.

Cluster data are publicly available via the Cluster Science Archive at https://csa.esac.esa.int/csa-web/, including the Cluster FGM data set
at https://doi.org/10.5270/esa-hxcrsz5.

We acknowledge Thilo Glißmann's work to provide a preliminary list of solar wind intervals for the solar wind data set.

We also want to thank Aris Valavanoglou at the Space Research Institute of the Austrian Academy of Sciences in Graz, Austria, for
providing the AMR data set used in this work.

Additionally, we want to thank Dragos Constantinescu at the Institute of Geophysics and Extraterrestrial Physics, TU Braunschweig in
Braunschweig, Germany, for his valuable insights on the calibration and cleaning of magnetometer data.

ChatGPT was used to improve the language of individual sentences in the final paper version.



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
