# Peer review of "Comparison of noise levels of two magnetometer types and their suitability for different space environments"

_EGUsphere, 2025_

## Referee Comment (RC1)

**"General comments"**

The paper "Comparison of noise levels of different magnetometer types and space environments" by Gerlinde Timmermann et al. contributes to the instrumental base for studying magnetic phenomena in various areas of space. The scientific and technical questions addressed in the paper are within the scope of GI.

The authors carefully and comprehensively thoroughly analyze the intensity and spectral characteristics of low-frequency (from 0.001 to 2 Hz) magnetic field oscillations in various regions around Earth and evaluates the applicability of two types of magnetometers (fluxgate (FGM) and anisotropic magnetoresistive (AMR)) for measuring such signals. The noise performance of an instrument is used as the primary characteristic for comparison. As a FGM instrument a science grade magnetometer similar to those used on Rosetta, THEMIS and JUICE space missions was selected for the noise level estimation. One of the instruments of the magnetic measurement system, known as the Service Oriented Space Magnetometer (SOSMAG), was chosen as an AMR type magnetometer. This device was installed inside the Korean GEO KOMPSAT–2A satellite to correct high-intensity magnetic interference generated inside the satellite in the recordings of more precise fluxgate magnetometers in the SOSMAG system.

A comparison of noise level measurements shows a significant advantage of the FGM magnetometer over the AMR one. The authors concluded that the FGM instrument can be effectively used in all four scenarios considered for measuring magnetic phenomena in near-Earth space. The AMR magnetometer is capable of measuring natural signals only in the magnetosheath, where the most intensive magnetic field disturbances occur.

The paper does not answer the question of how the noise characteristics of the devices selected for analysis compare with various other FGM and AMR magnetometers in terms of this parameter. So it is not comparison of "different magnetometer types" as the paper title declares, but rather a case study of the two FGM and AMR instruments for space applications.

In general noise level of fluxgate and search coil sensors depends on physical dimensions of their magnetic cores: the larger the core, the lower the noise level. How do the dimensions of the sensors in question compare?

What will the noise level of the FGM instrument be if its sensor is reduced to the size of the AMR sensor? Or what will the noise level of the AMR magnetometer be if its sensor is increased to the size of the FGM sensor?

The authors based their research on analysis of a large number of related studies and this is reflected in the list of references.

The abstract clearly and completely represent the contents of the paper. The paper's title suggests a broader scope than what's actually covered in the paper.

The overall presentation of the research results is well structured, but some statements and formulations are not sufficiently clear. Abbreviations, symbols and units are fairly defined and used.

**page 4, lines 111-112**

"Additionally, this leads to a noise floor where all disturbances are from the spacecraft (as seen later) and only general space phenomena are perceptible."

In my opinion, this sentence is not clear enough. It would be helpful to explain its meaning in more detail and provide a link to a specific subsection (or subsections) in the article instead of "as seen later".

**page 4, lines 118-119**

"For each data set and frequency, the 5th, 10th, 25th, 50th, 75th, 90th, and 95th percentiles of the square roots of the PSDs  $(P(f)^{1/2}$ , also called the amplitude spectral density) are calculated."

Please explain the rule used to calculate the 5th, 10th, 25th, 50th, 75th, 90th, and 95th percentiles of the PSD square roots.

**page 11, lines 228-230**

"The amplitude spectral density of the FGM is shown in Fig. 1(a). It has a spectral slope of  $\alpha = -0.75$ , approximately following a 1/f noise spectrum as is expected in this frequency range (Hooge et al., 1981). It exhibits the lowest values of all data sets, going down to 9 pT Hz-1/2 at 1 Hz."

It appears that the article (Hooge et al., 1981) does not directly discuss 1/f noise in fluxgate magnetic sensors. At least, the chatGPT5 model provides the following response:

"That classic review (Hooge, Kleinpenning & Vandamme, Rep. Prog. Phys. 44, 479–532, 1981; DOI: 10.1088/0034-4885/44/5/001) surveys 1/f (flicker) noise across many electronic materials and devices (e.g., metals, semiconductors, MOS devices), but I can't find any indication that it treats fluxgate magnetic sensors specifically. Abstracts/records describe scope limited to general mechanisms and device phenomenology (e.g., McWhorter vs. mobility-fluctuation pictures) rather than fluxgates."

It would be helpful to indicate which section of the review (Hooge et al., 1981), if any, refers to magnetic noise in fluxgate sensors.

The topic of FGM magnetic noise (MN) is discussed in the paper Korepanov et al., 2001, see Section 2, p. 138:

"By its character MN is similar to flicker noise, for which  $\alpha$ =1 (and even more than 1) is usually taken. However our detailed research of the MN frequency spectrum for different materials and designs in the frequency band up to 1 mHz showed that the value of  $\alpha$  for FGS has to be accepted in the limits of 0.75–0.8."

Please note that Korepanov et al.,2001, used the slope parameter  $\alpha$  in the formula  $b_F(f) = b_{F0} [1 + (f_0/f)^{\alpha}]$ , where  $b_{F0}$  is the minimum  $b_F(f)$  value at relatively high frequencies, and  $f_0$  is the corner frequency. Thus, their positive value of the parameter  $\alpha$  corresponds to the negative value of the slope  $\alpha$  in the paper under consideration.

**"Technical corrections"**

**page 5, Table 1 and lines 132-134, page 11, line 237**

The values of the amplitude spectral density of FGM and AMR magnetometers at 1 mHz (77.91 and 1277.36) and AMR at 1 Hz (157.09) in Table1 are different in comparison with values at these frequencies on the plots in Fig.1, 2, A1, A2 (110-115 at 1mHz for FGM, 2100 at 1 mHz and 210 at 1 Hz for AMR). It appears that the values (77.91 and 1277.36) in column  $P(f)^{1/2}$  @ 3 mHz (1 mHz) of Table 1 are actually given at 3 mHz, not 1 mHz as stated in the Table 1 capture and line 134. The noise level of AMR magnetometer also mentioned later (page 11, line 237) as 157 pT/Hz1/2 at 0.5 Hz is also in contradiction with plots in Figures 1(b), 2, A1, A2.

Neither 157, nor 210 pT/Hz1/2 at 1 Hz does not corresponds to the overall instrument noise density 100 pT/Hz1/2 at 1 Hz specified in Table IV "SOSMAG AMR Instrument Spec" of the paper (Leitner et al., 2015).

In addition, all amplitude spectral density values in Table 1 are unnecessarily precise. In my opinion, these numbers can be rounded to 1 pT without any loss of information.

**page 11, lines 236-239**

"However, the general noise level is significantly higher with  $157 \, \text{pT Hz}^{-1/2}$  at  $0.5 \, \text{Hz}$  compared to the fluxgate magnetometer as already seen in the last section. Since this is at quite a high level, this AMR magnetometer is not suitable for scientific measurements at lower frequencies though there are other AMRs with lower noise floors as described in Brown et al. (2012)"

What means expression "at lower frequencies"? Lower than what and why "not suitable"? Taking into account spectral slopes of the natural phenomena (-1.81 ... -1.36 at f<0.1 Hz) and that of the AMR magnetometer ( $\alpha$ =-1.02 at f<0.03) the lower frequency, the higher signal/noise ratio is expected. Thus, at a sufficiently low frequency, the signal-to-noise ratio becomes suitable for measuring natural signals, doesn't it?

**References**

Korepanov, V., Berkman, R., Rakhlin, L., Klymovych, Y., Prystai, A., Marussenkov, A., & Afanassenko, M. (2001). Advanced field magnetometers comparative study. *Measurement*, 29(2), 137-146.

---

## Author Comment (AC3)

**Answers to the comments from Anonymous Referee #1**

We thank the referee very much for their time and their detailed comments! We are sure the proposed changes will improve the paper. In the following, we will address each point separately.

**General comments:**

"The paper does not answer the question of how the noise characteristics of the devices selected for analysis compare with various other FGM and AMR magnetometers in terms of this parameter. So it is not comparison of "different magnetometer types" as the paper title declares, but rather a case study of the two FGM and AMR instruments for space applications."

"The paper's title suggests a broader scope than what's actually covered in the paper."

The title was changed to "Comparison of noise levels of two magnetometer types and their suitability for different space environments" to represent we only used two magnetometers and different space regions. Additionally, we made it clearer that the FGM used is state-of-the-art and thus comparable to others (page 3, line 80).

"In general noise level of fluxgate and search coil sensors depends on physical dimensions of their magnetic cores: the larger the core, the lower the noise level. How do the dimensions of the sensors in question compare?"

The size of the ring cores of the FGMs used is 13 mm (for XZ direction) and 18 mm (for YZ direction) (analog to the ring cores used in Auster et al., 2008). This was also added in the paper (line 81f.)

The AMR hybrid sensor has a volume of 1.67 cm3 with a 16 mm diameter and 8.3 mm height. (Leitner et al., 2015) This was also added in the paper (line 87).

"What will the noise level of the FGM instrument be if its sensor is reduced to the size of the AMR sensor? Or what will the noise level of the AMR magnetometer be if its sensor is increased to the size of the FGM sensor?"

It is of course true that there is a correlation between the general noise level of an FGM instrument and the physical dimensions of its magnetic core. Of comparable influence are the impacts on the noise level due to production variation, excitation power, feedback noise, winding numbers or pickup tuning, to name a few. In addition, bigger fluxgates are heavier and need more power, making them disadvantaged for use in space applications.

Nevertheless, this study was not targeting a full parameter study of fluxgate sensors, but to illustrate the usability of the presented instruments for different space environments.

It is a valid point to say the AMR and the FGM used are of different sizes. AMR magnetometers are smaller than FGMs and need less power. However, they are based on thin films in the micrometer range that give them their uniqe properties. Increasing the AMR to the size of the FGM would make them lose the thin film specific desired behavior, so they would not yield the same functionality as before. Additionally, these components would not be off-the-shelf available anymore and thus increase the price, which is a key selling point for using AMR magnetometers in the first place (low cost).

There is work being done on so-called fluxgate on chip, see e.g. Ma et al., 2024 (600  $\mu$ m ring core) or Lu et al., 2014 (less than 1 mm wide core). These come with higher noise levels than the fluxgate sensor used in this paper (61-87 pT/Hz1/2 at 1 Hz depending on direction in Ma et al., 2024; 50 pT/Hz1/2 @ 1 Hz with 50 Hz excitation and 790 pT/Hz1/2 @ 1 Hz with 25 Hz excitation in Lu et al., 2014), but they are mostly still lower than the noise levels of the AMR magnetometer presented in this study. Furthermore, we think going into such detail of the different sizes of magnetometers is – though quite interesting – out of the scope of this paper.

"The overall presentation of the research results is well structured, but some statements and formulations are not sufficiently clear. Abbreviations, symbols and units are fairly defined and used."

We checked again for consistency in abbreviations, symbol usage, and units and made corrections/adjustments where necessary.

**Specific comments:**

**page 4, lines 112-114:**

"In my opinion, this sentence is not clear enough. It would be helpful to explain its meaning in more detail and provide a link to a specific subsection (or subsections) in the article instead of "as seen later"."

**The sentence was changed and a link to the results section is now given.**

Additionally, this lets small variations of individual measurements fade into the background noise level, so all disturbances seen are either from the spacecraft or a result of general space phenomena (as seen in section 3).

**page 4, lines 121-123:**

"Please explain the rule used to calculate the 5th, 10th, 25th, 50th, 75th, 90th, and 95th percentiles of the PSD square roots."

**An explanation for the percentiles is now included in the text.**

Here, the 5th percentile corresponds to the value that is equal to 5% of the maximum value at each frequency, the 10th percentile corresponds to the value that is equal to 10% of the maximum value at each frequency and so forth.

**page 11, lines 232-234:**

"It appears that the article (Hooge et al., 1981) does not directly discuss 1/f noise in fluxgate magnetic sensors. (...) The topic of FGM magnetic noise (MN) is discussed in the paper Korepanov et al., 2001, see Section 2, p. 138: (...)"

Thank you very much for pointing out a better source for a comparison value of spectral slope! The old one has been replaced.

**Technical Corrections:**

page 5, Table 1 and lines 136-138, page 12, line 242:

"The values of the amplitude spectral density of FGM and AMR magnetometers at 1 mHz (77.91 and 1277.36) and AMR at 1 Hz (157.09) in Table1 are different in comparison with values at these frequencies on the plots in Fig.1, 2, A1, A2 (110-115 at 1mHz for FGM, 2100 at 1 mHz and 210 at 1 Hz for AMR). It appears that the values (77.91 and 1277.36) in column

 $P(f)_{1/2}$  @ 3 mHz (1 mHz) of Table 1 are actually given at 3 mHz, not 1 mHz as stated in the Table 1 capture and line 134. The noise level of AMR magnetometer also mentioned later (page 11, line 237) as 157 pT/Hz1/2 at 0.5 Hz is also in contradiction with plots in Figures 1(b), 2, A1, A2."

The unmatching values of Table I and the caption/text have been corrected.

"Neither 157, nor 210 pT/Hz1/2 at 1 Hz does not corresponds to the overall instrument noise density 100 pT/Hz1/2 at 1 Hz specified in Table IV "SOSMAG AMR Instrument Spec" of the paper (Leitner et al., 2015)."

I got the measurement data from A. Valavanoglou (as shown in the reference list), if plotted one can see it is not the exact same measurement as used in Leitner et al. (2015) (difference in levels of B and temperature). Nevertheless, they are from sensors developed for SOSMAG. The noise levels of 210 pT/Hz1/2 are in the same order of magnitude as given in the paper (100 pT/Hz1/2), thus saying "an AMR magnetometer as used in SOSMAG" seems - in our opinion - appropriate.

"In addition, all amplitude spectral density values in Table 1 are unnecessarily precise. In my opinion, these numbers can be rounded to 1 pT without any loss of information."

The values have been rounded to 1 pT.

page 11, lines 236-239:

"What means expression "at lower frequencies"? Lower than what and why "not suitable"?"

The expression "at lower frequencies" has been specified as well as the meaning of "not suitable". Since this is at quite a high level, this AMR magnetometer is not suitable for scientific measurements at frequencies below 1 Hz, since the noise floor is higher than the lowest expected signals. There exist other AMRs with lower noise floors as described in Brown et al. (2012).

"Taking into account spectral slopes of the natural phenomena (-1.81 ... -1.36 at f<0.1 Hz) and that of the AMR magnetometer ( $\alpha$ =-1.02 at f<0.03) the lower frequency, the higher signal/noise ratio is expected. Thus, at a sufficiently low frequency, the signal-to-noise ratio becomes suitable for measuring natural signals, doesn't it?"

We agree that in theory, at sufficiently low frequencies the SNR of an AMR should become suitable for measuring natural signals. However, the lowest frequency is given by the length of the time interval measured. For this study, the time intervals of the natural signal measurements were chosen to be 5 min long, giving a minimum frequency of 3.3 mHz. Also, it needs to be taken into account that the length of the time intervals is restricted by how long a spacecraft can dwell in the space region under consideration. This is restricted by the spacecraft's velocity, which is usually in the one digit km/s range, given e.g. for a THEMIS satellite an orbit period of about 26 h. In this time period, it will cross several space regions. If it was staying in only one space region, this would yield frequencies in the  $\mu$ Hz range. At last, the offset stability for AMR magnetometer cannot be assumed to be constant for long time periods. (see Schulz et al., 2019)

**References**

Auster, H. U., Glassmeier, K. H., Magnes, W., Aydogar, O., Baumjohann, W., Constantinescu, D., Fischer, D., Fornacon, K. H., Georgescu, E., Harvey, P., Hillenmaier, O., Kroth, R., Ludlam, M., Narita, Y., Nakamura, R., Okrafka, K., Plaschke, F., Richter, I., Schwarzl, H., Stoll, B., Valavanoglou, A., and Wiedemann, M. (2008). The THEMIS Fluxgate Magnetometer. Space Science Review (2008) 141: 235–264.

Leitner, S., Valavanoglou, A., Brown, P., Hagen, C., Magnes, W., Whiteside, B. J., Carr, C. M., Delva, M., and Baumjohann, W. (2015). Design of the Magnetoresistive Magnetometer for ESA's SOSMAG Project. IEEE Transactions on Magnetics, Vol. 51, No. 1, January 2015, 4001404.

Lu, C-C., Huang, J., Chiu, P-K., Chiu, S-L., and Jeng, J-T. (2014). High-Sensitivity Low-Noise Miniature Fluxgate Magnetometers Using a Flip Chip Conceptual Design. Sensors 2014, 14, 13815-13829.

Ma, Q., Dai, Y., Wu, T., Chen, H., Sun, X., Lei, C. (2024). A multidimensional integrated micro three-component fluxgate sensor based on microelectromechanical system technology. Sensors and Actuators: A. Physical, 371 (2024) 115315.

Schulz, L., Heinisch, P., and Richter, I. (2019). Calibration of Off-the-Shelf Anisotropic Magnetoresistance Magnetometers. Sensors 2019, 19, 1850.